# PCSK9 Inhibitor Inclisiran Attenuates Cardiotoxicity Induced by Sequential Anthracycline and Trastuzumab Exposure via NLRP3 and MyD88 Pathway Inhibition

**DOI:** 10.3390/ijms26146617

**Published:** 2025-07-10

**Authors:** Vincenzo Quagliariello, Massimiliano Berretta, Irma Bisceglia, Martina Iovine, Matteo Barbato, Raffaele Arianna, Maria Laura Canale, Andrea Paccone, Alessandro Inno, Marino Scherillo, Stefano Oliva, Christian Cadeddu Dessalvi, Alfredo Mauriello, Carlo Maurea, Celeste Fonderico, Anna Chiara Maratea, Domenico Gabrielli, Nicola Maurea

**Affiliations:** 1Division of Cardiology, Istituto Nazionale Tumori-IRCCS-Fondazione G. Pascale, 80131 Napoli, Italy; mart.iovine@gmail.com (M.I.); matteo.barbato@istitutotumori.na.it (M.B.); raffaele.arianna@istitutotumori.na.it (R.A.); andrea.paccone@istitutotumori.na.it (A.P.); alfredo.mauriello@istitutotumori.na.it (A.M.); celeste.fonderico@istitutotumori.na.it (C.F.); annachiara.maratea@istitutotumori.na.it (A.C.M.); n.maurea@istitutotumori.na.it (N.M.); 2Department of Clinical and Experimental Medicine, University of Messina, 98125 Messina, Italy; berrettama@gmail.com; 3Servizi Cardiologici Integrati, Dipartimento Cardio-Toraco-Vascolare, Azienda Ospedaliera San Camillo Forlanini, 00152 Roma, Italy; irmabisceglia@gmail.com; 4U.O.C. Cardiologia, Ospedale Versilia, 55041 Lido di Camaiore, Italy; marialaura.canale@uslnordovest.toscana.it; 5Medical Oncology, IRCCS Ospedale Sacro Cuore Don Calabria, 37024 Negrar di Valpolicella, Italy; alessandro.inno@sacrocuore.it; 6Cardiologia Interventistica e UTIC, A.O. San Pio, Presidio Ospedaliero Gaetano Rummo, 82100 Benevento, Italy; marino.scherillo@libero.it; 7Cardio-Oncology Unit, IRCCS Istituto Tumori, “Giovanni Paolo II”, 70124 Bari, Italy; s.oliva@oncologico.bari.it; 8Department of Medical Sciences and Public Health, University of Cagliari, 09124 Cagliari, Italy; cadedduc@unica.it; 9UOC Neurology-Stroke Unit, AORN Cardarelli, 80131 Naples, Italy; carlo.maurea@libero.it; 10UOC Cardiologia, Dipartimento Cardio-Toraco-Vascolare, Azienda Ospedaliera San Camillo Forlanini, Roma—Fondazione per il Tuo Cuore—Heart Care Foundation, 00152 Firenze, Italy; dgabrielli@scamilloforlanini.rm.it

**Keywords:** cancer, metabolism, LDL, NLRP3, cardio-oncology, cardiotoxicity, PCSK9

## Abstract

Cardiotoxicity related to anthracyclines and trastuzumab represents a significant clinical challenge in cancer therapy, often limiting treatment efficacy and patient survival. The underlying mechanisms of cardiotoxicity involve the activation of NLRP3 and the MyD88-dependent signaling pathway. Proprotein convertase subtilisin/kexin type 9 inhibitors (PCSK9i), such as inclisiran, are known for their lipid-lowering effects, but emerging data indicate that they may also exert pleiotropic benefits beyond cholesterol reduction. This study investigates whether inclisiran can mitigate the cardiotoxic effects of anthracyclines and trastuzumab through reduction of NLRP3 activation and MyD88 signaling, independently of its effects on dyslipidemia. Human induced pluripotent stem cell-derived cardiomyocytes (hiPSC-CMs) were exposed to subclinical concentrations of doxorubicin (1 µM) and trastuzumab in sequential therapy (200 nM), alone or in combination with inclisiran (100 nM) for 24 h. After the incubation period, we performed the following tests: determination of cardiomyocytes apoptosis, analysis of intracellular reactive oxygen species, lipid peroxidation products (including malondialdehyde and 4-hydroxynonenal), intracellular mitofusin-2 and Ca^++^ levels. Troponin and BNP were quantified through selective ELISA methods. A confocal laser scanning microscope was used to study cardiomyocyte morphology and F-actin staining after treatments. Moreover, pro-inflammatory studies were also performed, including the intracellular expression of NLRP-3, MyD-88 and twelve cytokines/growth factors involved in cardiotoxicity (IL-1α, IL-1β, IL-2, IL-4, IL-6, IL-10, IL-12, IL17-α, IFN-γ, TNF-α, G-CSF, GM-CSF). Inclisiran co-incubated with doxorubicin and trastuzumab exerts significant cardioprotective effects, enhancing cell viability by 88.9% compared to only DOXO/TRA treated cells (*p* < 0.001 for all). Significant reduction of oxidative stress, and intracellular levels of NLRP-3, MyD88, IL-1α, IL-1β, IL-6, IL-12, IL17-α, TNF-α, G-CSF were seen in the inclisiran group vs. only DOXO/TRA (*p* < 0.001). For the first time, PCSK9i inclisiran has been shown to exert significant anti-inflammatory effects to reduce anthracycline-HER-2 blocking agent-mediated cardiotoxicity through NLRP-3 and Myd-88 related pathways. The overall conclusions of the study warrant further investigation of the use of PCSK9i in primary prevention of CTRCD in cancer patients, independently from dyslipidemia.

## 1. Introduction

Cardiotoxicity remains a significant challenge in cancer therapy, particularly in patients receiving sequential treatment with doxorubicin and trastuzumab [1]. Doxorubicin, an anthracycline widely used in chemotherapy, is associated with dose-dependent cardiotoxic effects, primarily mediated through oxidative stress, mitochondrial dysfunction, and apoptosis in cardiomyocytes [2,3]. The cumulative damage induced by doxorubicin can lead to irreversible cardiac dysfunction, increasing the risk of heart failure in cancer survivors. Trastuzumab, a monoclonal antibody targeting HER2-positive breast cancer, further exacerbates cardiotoxicity when administered sequentially after doxorubicin [4,5]. Unlike doxorubicin, trastuzumab-induced cardiotoxicity is largely reversible and is believed to arise from impaired HER2 signaling, which is crucial for cardiomyocyte survival. Nevertheless, the combined administration of these agents significantly increases the likelihood of left ventricular dysfunction, limiting their clinical use and necessitating effective cardioprotective strategies [6,7]. According to the 2022 ESC Guidelines on Cardio-Oncology, the risk of cancer therapy-related cardiac dysfunction (CTRCD) is significantly elevated in patients receiving sequential anthracycline followed by trastuzumab treatment, with reported incidence rates of CTRCD reaching 27–32% in this population. This clinical scenario therefore represents a relevant and urgent context in which to explore potential cardioprotective strategies. Inclisiran, a small interfering RNA (siRNA) targeting proprotein convertase subtilisin/kexin type 9 (PCSK9), has emerged as a potent lipid-lowering agent with a favorable safety profile [8,9]. By reducing PCSK9 levels, inclisiran enhances low-density lipoprotein receptor (LDLR) recycling, leading to sustained reductions in circulating cholesterol levels [10]. While its primary use is in managing hypercholesterolemia, inclisiran has also demonstrated pleiotropic effects, including anti-inflammatory and endothelial-protective properties, which may confer cardiovascular benefits beyond lipid lowering [11]. Recent evidence suggests that inclisiran may exert protective effects in both cancer and non-cancer patients [12]. In oncology, PCSK9 inhibition has been linked to improved immune surveillance and potential tumor-suppressive effects, making inclisiran an attractive candidate for adjunctive therapy in cancer patients [13]. Moreover, its cardiovascular benefits extend to patients without overt dyslipidemia, suggesting that inclisiran’s mechanisms of action may be relevant in mitigating chemotherapy-induced cardiotoxicity [14]. Emerging interest has focused on the role of PCSK9 not only in lipid metabolism but also in modulating inflammation, oxidative stress, and cellular apoptosis, key mechanisms implicated in chemotherapy-induced cardiotoxicity. Although PCSK9 inhibitors such as inclisiran have been shown to reduce LDL-C and improve cardiovascular outcomes in high-risk patients (e.g., in ORION trials), their utility in oncology settings remains unexplored in clinical trials [13]. There are no published studies investigating PCSK9 inhibition during chemotherapy in human patients. Nonetheless, preclinical data, including studies using monoclonal antibody-based PCSK9 inhibitors, demonstrate protection against anthracycline- and trastuzumab-induced cardiotoxicity via anti-inflammatory and antioxidative mechanisms [14]. Additionally, doxorubicin has been shown to upregulate PCSK9 expression in myocardial tissue, further supporting a potential link between PCSK9 activity and chemotherapy-induced cardiac injury. Taken together, these findings suggest a mechanistic rationale for evaluating PCSK9 inhibitors as cardioprotective agents in cancer therapy, forming the basis for the present study. Investigating its efficacy in this context may provide a novel strategy for preserving cardiac function in cancer patients undergoing sequential anthracycline and HER2-targeted therapies, thereby improving long-term cardiovascular outcomes in this vulnerable population.

## 2. Results

### 2.1. Cardioprotective Properties of Inclisiran During Anthracyclines and Trastuzumab Therapy in Sequential Regimen

Figure 1 illustrates the cardioprotective efficacy of PCSK9 inhibition via inclisiran during sequential treatment with doxorubicin and anti-HER2 monoclonal antibodies. Cell viability was evaluated across four treatment groups in hiPSC-derived cardiomyocytes. As shown in Figure 1A, exposure to doxorubicin combined with anti-HER2 monoclonal antibodies (DOXO + Anti-HER2 mAb) significantly reduced cell viability to approximately 45% compared to untreated controls (*p* < 0.001), confirming the known cardiotoxic effect of this combination. Treatment with PCSK9 inhibitor (PCSK9i; inclisiran) alone showed no cytotoxicity, maintaining viability close to 100%, not significantly different from the untreated group (*p* > 0.05). Remarkably, cells co-treated with PCSK9i and DOXO + Anti-HER2 mAb exhibited a significant improvement in viability, reaching approximately 85%, compared to the DOXO + Anti-HER2 mAb group alone (*p* < 0.001). This suggests a cardioprotective effect of PCSK9 inhibition against anthracycline/anti-HER2-induced toxicity. The rescue effect was also statistically significant compared to PCSK9i alone (*p* < 0.01), indicating a specific interaction in the presence of cardiotoxic stress. Moreover, caspase-3 activity (Figure 1B) was markedly increased in the DOXO + Anti-HER2 mAb group compared to untreated control cells (*p* < 0.001), indicating robust apoptotic activation. Similarly, myocardial lactate dehydrogenase (LDH) release (Figure 1C) and cytochrome C expression (Figure 1D) were significantly elevated in response to doxorubicin and trastuzumab, reflecting both cell membrane disruption and mitochondrial damage (*p* < 0.001 for both).

Inclisiran monotherapy (PCSK9i) did not significantly alter caspase-3, LDH, or cytochrome C levels compared to untreated cells, suggesting no inherent cytotoxicity. Importantly, treatment with inclisiran in the PCSK9i + DOXO + Anti-HER2 mAb group significantly attenuated the increase in these markers compared to the DOXO + Anti-HER2 mAb group alone (*p* < 0.001 for all comparisons), demonstrating a strong cytoprotective and anti-apoptotic effect. In Figure 1E, intracellular calcium levels were significantly elevated following DOXO + Anti-HER2 mAb treatment (*p* < 0.001 vs. control), consistent with calcium dysregulation observed during cardiotoxic stress. Inclisiran significantly reduced this calcium overload (*p* < 0.001 vs. DOXO + Anti-HER2 mAb), supporting its role in maintaining calcium homeostasis under chemotherapeutic stress.

Finally, as shown in Figure 1F, mitofusin-2 levels, a key regulator of mitochondrial fusion and integrity, were significantly reduced following DOXO + Anti-HER2 mAb exposure (*p* < 0.001 vs. control), indicating mitochondrial dysfunction. Inclisiran co-treatment restored mitofusin-2 expression to near basal levels (*p* < 0.001 vs. DOXO + Anti-HER2 mAb), suggesting that PCSK9 inhibition preserves mitochondrial structure and function during cardiotoxic chemotherapy. Collectively, these data support a protective role of PCSK9 inhibition in preventing chemotherapy-induced cardiotoxicity by reducing apoptosis, preserving mitochondrial integrity, and restoring calcium homeostasis.

### 2.2. Effects of Inclisiran on Intracellular Reactive Oxygen Species and Lipid Peroxidation Levels During Exposure to Anthracyclines and Trastuzumab

Figure 2 demonstrates the potent antioxidative properties of PCSK9 inhibition with inclisiran in cardiomyocytes exposed to sequential anthracycline and trastuzumab therapy. Panel A shows that treatment with doxorubicin and anti-HER2 monoclonal antibodies (DOXO + Anti-HER2 mAb) significantly elevated intracellular reactive oxygen species (iROS) in HIPSC-CMS cardiomyocytes, as indicated by a marked increase in fluorescence intensity compared to untreated control cells (*p* < 0.001). This oxidative stress response was significantly mitigated by inclisiran co-treatment (PCSK9i + DOXO + Anti-HER2 mAb), with iROS levels reduced by over 60% relative to the DOXO + Anti-HER2 mAb group (*p* < 0.001). Notably, inclisiran alone did not induce oxidative stress, and iROS levels remained comparable to control (*p* = 0.002 vs. DOXO + Anti-HER2 mAb), confirming its safety profile in cardiac cells. Panel B further explores lipid peroxidation markers, revealing that both malondialdehyde (MDA) and 4-hydroxy-2-nonenal (4-HNA) were significantly increased following DOXO + Anti-HER2 mAb exposure (*p* < 0.001 for both vs. control). These markers are indicative of membrane lipid oxidative degradation and are commonly associated with cardiotoxicity. Inclisiran monotherapy maintained MDA and 4-HNA levels close to those observed in untreated cells. Crucially, co-treatment with inclisiran significantly suppressed the DOXO + Anti-HER2-induced elevation in MDA (*p* < 0.001) and 4-HNA (*p* = 0.0073), underscoring its efficacy in counteracting lipid peroxidation. Collectively, these findings underscore that PCSK9 inhibition by inclisiran confers robust antioxidative protection in cardiomyocytes subjected to chemotherapeutic stress. By significantly reducing intracellular ROS production and downstream lipid peroxidation products, inclisiran may serve as a valuable cardioprotective agent during anthracycline–trastuzumab therapy.

### 2.3. Biomarkers of Heart Failure: Analysis of H-FABP, Troponin-T and BNP

Figure 3 illustrates the myocardial protective role of inclisiran through attenuation of cardiac injury biomarkers in HIPSC-CMS cardiomyocytes exposed to doxorubicin and trastuzumab administered in a sequential regimen. As shown in Figure 3A, Heart-type Fatty Acid Binding Protein (H-FABP) levels were markedly increased to approximately 10 ng/mL in the DOXO + Anti-HER2 mAb group compared to baseline levels of ~2.5 ng/mL in untreated cells (*p* < 0.001). This indicates substantial early myocardial injury. PCSK9 inhibition with inclisiran alone did not elevate H-FABP. Notably, co-treatment with inclisiran significantly decreased H-FABP to 3.75 ng/mL, corresponding to a 62.5% reduction relative to the DOXO + Anti-HER2 mAb group (*p* < 0.001), underscoring its potent protective effect against anthracycline/trastuzumab-induced myocardial stress. Figure 3B reports Troponin T levels, a sensitive and specific marker of cardiomyocyte necrosis. Exposure to DOXO + Anti-HER2 mAb led to a sharp elevation to 0.95 ng/mL (*p* < 0.001 vs. control), whereas inclisiran co-treatment significantly reduced this to 0.22 ng/mL, representing a 76.8% reduction (*p* < 0.001 vs. DOXO + Anti-HER2 mAb). This suggests that inclisiran effectively preserves myocardial membrane integrity under chemotherapeutic insult. Moreover, Figure 3C shows B-type Natriuretic Peptide (BNP) levels, a clinical marker of ventricular wall stress and early heart failure. BNP increased to 248 pg/mL in the DOXO + Anti-HER2 mAb group (*p* < 0.01 vs. control). Inclisiran co-treatment significantly lowered BNP to 148 pg/mL, reflecting a 40.3% reduction compared to the DOXO + Anti-HER2 mAb group (*p* < 0.05), indicating alleviation of cardiac load and improved functional status.

### 2.4. Effect of Inclisiran on NLRP-3, MyD-88 and Cytokines During Exposure to Anthracyclines and Trastuzumab Therapy in Sequential Regimen

Figure 4 explores the anti-inflammatory potential of PCSK9 inhibition with inclisiran in cardiomyocytes subjected to anthracycline and trastuzumab-induced inflammatory injury. The figure focuses on intracellular expression of key inflammatory mediators including NLRP3, MyD88 and a broad panel of cytokines and chemokines. The expression of NLRP3 (Figure 4A), a central mediator of inflammasome activation, was significantly upregulated in cardiomyocytes treated with DOXO + Anti-HER2 mAb, reaching a ~12-fold increase over baseline (untreated control; *p* < 0.001). PCSK9i alone did not significantly alter NLRP3 expression. Co-treatment with PCSK9i during DOXO + Anti-HER2 exposure reduced NLRP3 levels to ~6-fold, representing a 50% reduction compared to the DOXO + Anti-HER2 mAb group (*p* < 0.001), indicating a strong anti-inflammasome effect. Expression of Myeloid Differentiation Primary Response 88 (MyD88), a key adaptor molecule in Toll-like receptor (TLR) signaling and inflammasome priming, was elevated to ~10.5-fold in the DOXO + Anti-HER2 mAb group (*p* < 0.001 vs. untreated) (Figure 4B). Inclisiran treatment in combination with DOXO + Anti-HER2 significantly decreased MyD88 levels to ~5.5-fold, representing a 47.6% reduction (*p* < 0.001). PCSK9i alone maintained MyD88 levels close to baseline (~2.5-fold), further confirming the non-inflammatory profile of inclisiran in the absence of chemotherapeutic stress. In Figure 4C, cytokine analysis demonstrated that DOXO + Anti-HER2 mAb treatment elicited a strong pro-inflammatory response, characterized by significant upregulation of key cytokines such as IL-1α, IL-1β, IL-6, TNF-α, G-CSF, GM-CSF, IL-12, IL-17A, and IL-18. Co-treatment with inclisiran markedly attenuated these elevations, with most cytokines showing reductions of approximately 40–50%, indicating effective suppression of the inflammatory cascade. In contrast, IL-2 and IL-4 levels remained unchanged across treatment groups. Importantly, IL-10—a major anti-inflammatory cytokine—was selectively increased in the inclisiran co-treatment group, suggesting a shift toward a cardioprotective immune profile. These findings support inclisiran’s immunomodulatory effects, not only through inflammasome inhibition but also via promotion of anti-inflammatory signaling pathways.

### 2.5. Confocal Laser Scanning Microscope (CLSM) Analysis of F-Actin Expression and Morphology in Cardiomyocytes

Anthracyclines and HER2-targeted agents are well-documented to induce cytoskeletal disruption in cardiomyocytes, contributing to impaired contractility and structural integrity of the myocardium [15]. Given the central role of PCSK9 in lipid regulatory pathways and membrane remodeling, we investigated whether inclisiran could preserve F-actin integrity in human iPSC-derived cardiomyocytes exposed to sequential doxorubicin and trastuzumab treatment. Imaging results clearly show that F-actin is highly expressed in untreated cardiac cells (Figure 5A) while anthracyclines and trastuzumab reduced significantly its expression and localization (Figure 5B). In fact, we can see that F-actin loses its linear expression in the cell cytoplasm, accumulating in perinuclear areas with reduced fluorescent staining, indicative of the reduced cytoskeletal architecture. Moreover, inclisiran in monotherapy did not change F-actin expression in cardiomyocytes, confirming its safety in these cells (Figure 5C). Notably, treatment with inclisiran preserves cytoskeletal de-structuring in the myocyte (Figure 5D), showing a more linear staining with a markedly more conserved intracellular architecture compared to the Doxo-Tra group. Spectrofluorimetric quantification confirmed significant differences in F-actin integrity across treatment conditions (Figure 5E). Cells treated with DOXO + Anti-HER2 showed a marked reduction in F-actin fluorescence intensity (~52% decrease vs. control, *p* < 0.001), indicating cytoskeletal degradation. In contrast, PCSK9i alone did not significantly affect fluorescence intensity (+4% vs. control, *p* = ns). Importantly, co-treatment with PCSK9i restored F-actin levels, showing only a ~12% reduction compared to control (*p* < 0.05 vs. control; *p* < 0.001 vs. DOXO + Anti-HER2). These findings support the protective effect of PCSK9 inhibition in preserving cytoskeletal structure under cardiotoxic stress.

## 3. Discussion

The findings of this study provide compelling evidence for the cardioprotective potential of inclisiran in reducing the cardiotoxic effects of doxorubicin and trastuzumab in sequential therapy. According to the 2022 European Society of Cardiology (ESC) Guidelines on Cardio-Oncology, patients with breast cancer undergoing sequential treatment with anthracyclines followed by trastuzumab represent one of the highest-risk populations for developing CTRCD, with reported incidence rates reaching 27–32% [16]. This elevated risk is attributable to the cumulative and synergistic cardiotoxic effects of these agents, whereby anthracyclines induce dose-dependent oxidative damage and mitochondrial dysfunction, while trastuzumab disrupts HER2-mediated cardiomyocyte survival pathways. The sequential administration of these therapies, although clinically effective against HER2-positive breast cancer, significantly compromises myocardial integrity, especially in patients with additional cardiovascular risk factors or subclinical cardiac vulnerability [16]. Therefore, the identification of safe and effective cardioprotective interventions is an urgent clinical priority. The current findings, demonstrating the cardioprotective potential of the PCSK9 inhibitor inclisiran through modulation of the NLRP3 inflammasome and MyD88 signaling pathways, are particularly relevant. The mechanistic underpinnings of doxorubicin-induced cardiotoxicity are well-documented, involving the generation of excessive reactive oxygen species (ROS) [16] that impair mitochondrial integrity, disrupt calcium homeostasis [17], and activate apoptotic pathways in cardiomyocytes [18]. This oxidative stress, coupled with NLRP3/IL-1 activation, triggers DNA damage and myocyte loss, contributing to cumulative cardiac dysfunction [19,20]. Additionally, trastuzumab-induced cardiotoxicity is primarily linked to HER2 pathway inhibition, which compromises cardiomyocyte survival by impairing essential prosurvival signaling cascades, including the phosphoinositide 3-kinase (PI3K)/Akt and extracellular signal-regulated kinase (ERK) pathways [21,22,23]. The epidemiological burden of chemotherapy-induced cardiotoxicity is substantial, with anthracycline-induced heart failure occurring in up to 26% of patients receiving cumulative doses exceeding 550 mg/m^2^ [24]. Furthermore, trastuzumab-related cardiac dysfunction affects approximately 10–20% of patients, necessitating treatment interruption in many cases [25]. Given the rising incidence of cancer and improved patient survival, long-term cardiovascular complications have become a critical concern, emphasizing the need for effective cardioprotective interventions [26]. Inclisiran’s mechanism of action extends beyond lipid regulation [27], as emerging evidence highlights its role in modulating inflammation [28], endothelial function [29], and immune response [30]. PCSK9 inhibition has been shown to enhance endothelial nitric oxide synthase (eNOS) activity [31], reducing vascular oxidative stress and promoting nitric oxide (NO)-mediated vasodilation [32]. Furthermore, PCSK9 downregulation has been associated with reduced NLRP3 inflammasome activation, a key mediator of cardiac inflammation, suggesting that inclisiran may attenuate myocardial injury through anti-inflammatory pathways [33,34]. These pleiotropic effects provide a plausible explanation for its observed cardioprotective benefits, independent of cholesterol lowering.

Our data demonstrate, for the first time, that inclisiran effectively reduces NLRP3/IL-1 and MYD88/CCL2 signaling pathways in cardiac cells exposed to doxorubicin and trastuzumab in sequential therapy. The NLRP3 inflammasome is a crucial regulator of inflammation [35], myocarditis [36], and atherosclerosis [37], and its activation leads to IL-1β secretion, promoting myocardial inflammation and fibrosis [38]. The overexpression of NLRP3/IL-1β signaling has been implicated in chemotherapy-induced cardiotoxicity, exacerbating oxidative damage and cardiac dysfunction [39]. Similarly, MYD88 plays a pivotal role in immune cell recruitment and inflammatory amplification within the myocardium [40]. By suppressing these inflammatory pathways, inclisiran may prevent cardiotoxic injury, thereby preserving cardiac structure and function.

Figure 6 provides a schematic summary of the proposed cardioprotective and anti-inflammatory mechanisms of inclisiran in cardiomyocytes during anthracycline and trastuzumab therapy. It illustrates how trastuzumab binding to HER2 receptors, combined with intracellular doxorubicin accumulation, initiates a cascade of deleterious effects, beginning with the transcriptional upregulation of pro-inflammatory and pro-apoptotic genes (e.g., NLRP3, MYD88, IL-1β) and leading to increased mitochondrial ROS production, lipid peroxidation, and ferroptosis [41]. Additionally, doxorubicin disrupts mitochondrial and calcium homeostasis, exacerbating myocardial stress. Inclisiran, acting via RNA interference to silence PCSK9 expression, is shown to intercept this pathogenic cascade by reducing the transcription and translation of inflammatory and oxidative mediators [42]. This leads to reduced inflammasome activation, improved mitochondrial integrity, decreased ROS generation, reduced lipid peroxidation, and ultimately the prevention of ferroptotic cell death [43,44]. Notably, inclisiran also promotes an anti-inflammatory shift, as evidenced by the upregulation of IL-10, further contributing to myocardial resilience. To provide a more detailed rationale for the selected cytokine panel, we focused on 12 inflammatory and immunomodulatory mediators known to participate in cardiac inflammation, remodeling, and injury, processes critically implicated in anthracycline- and trastuzumab-induced cardiotoxicity. These cytokines and growth factors were chosen for their established or emerging roles in cardiomyocyte dysfunction, oxidative stress amplification, and maladaptive immune responses, all of which contribute to CTRCD. Notably, IL-1α and IL-1β are central pro-inflammatory cytokines that play key roles in inflammasome activation and myocardial injury. Both are upregulated in response to oxidative stress and mitochondrial damage, particularly via NLRP3 activation, which is strongly implicated in anthracycline-induced cardiotoxicity. IL-1β has also been shown to exacerbate cardiac fibrosis and ventricular dysfunction [40]. While primarily known for T-cell activation, IL-2 may contribute to systemic inflammation in the cardiac microenvironment. Elevated IL-2 has been reported in doxorubicin-treated models and may potentiate immune cell recruitment into cardiac tissue. Moreover, as a Th2 cytokine, IL-4 has a dual role in cardiotoxicity; notably, it can exert anti-inflammatory effects in certain contexts, but it also promotes fibrotic remodeling when chronically elevated, contributing to pathological cardiac remodeling [41]. A key mediator of both acute and chronic inflammation, IL-6 is consistently elevated in models of doxorubicin cardiotoxicity and in patients with CTRCD. It enhances ROS generation and is involved in endothelial dysfunction, contributing to both systolic and diastolic impairment. Moreover, IL-10 was quantified; this anti-inflammatory cytokine serves as a compensatory response to injury, inhibiting the synthesis of pro-inflammatory cytokines such as IL-1β and TNF-α. Increased IL-10 may reflect an endogenous attempt to counteract chemotherapy-induced inflammation, and its modulation could reflect the efficacy of cardioprotective interventions. Moreover, we analyzed IL-12 (p40/p70) in cardiomyocytes; this heterodimeric cytokine bridges innate and adaptive immunity by promoting Th1 differentiation and IFN-γ production. Its overexpression is linked to cardiomyocyte apoptosis and adverse ventricular remodeling in inflammatory cardiac conditions [40,41]. Moreover, produced by Th17 cells, IL-17A has emerged as a potent contributor to cardiac inflammation, particularly by enhancing neutrophil recruitment and sustaining tissue-level oxidative stress. Its elevation has been observed in cardiac injury models, including doxorubicin exposure. A master regulator of cardiac inflammation, TNF-α is upregulated in response to anthracycline-induced mitochondrial stress and ROS production. It directly impairs myocardial contractility, promotes apoptosis, and contributes to left ventricular dysfunction. G-CSF (Granulocyte Colony-Stimulating Factor) has cardioprotective potential in certain contexts but may also reflect systemic inflammatory activation. Its role in CTRCD remains complex and context-dependent. Instead, GM-CSF (Granulocyte-Macrophage Colony-Stimulating Factor) modulates monocyte/macrophage activation and differentiation, contributing to sustained inflammation and cytokine release within the cardiac microenvironment [42,43,44]. In our study, treatment with inclisiran resulted in a significant reduction of several key pro-inflammatory cytokines, particularly IL-1β, IL-6, IL-17A, and TNF-α, aligning with its proposed inhibitory action on the NLRP3–MyD88 axis. This suppression of inflammatory signaling may underlie the observed attenuation of cardiomyocyte apoptosis, cytoskeletal disruption, and oxidative stress. By modulating this panel of cytokines, inclisiran appears to exert a broad anti-inflammatory and cardioprotective effect, supporting its potential role in preventing CTRCD in high-risk chemotherapy settings.

Figure 6 underscores the multifaceted role of inclisiran in attenuating the molecular drivers of cardiotoxicity at transcriptional, post-transcriptional, and mitochondrial levels. Notably, our findings suggest that inclisiran’s effects are independent of lipid levels, reinforcing its potential as a therapeutic agent for cardioprotection in patients at high risk of heart failure, especially those undergoing sequential anthracycline and HER2-targeted therapies [45,46]. This highlights the need for further investigations into its clinical applications, particularly through randomized controlled trials evaluating its long-term cardiovascular benefits in cancer patients. Despite the strength of these findings, several limitations must be acknowledged. First, this study was conducted exclusively in vitro using human cardiomyocyte-like HIPSC-CMS cells. While physiologically relevant, this model cannot fully reproduce the complex in vivo cardiac environment, including neurohormonal influences, systemic metabolism, and immune cell interactions [47]. Second, the lack of in vivo preclinical validation, such as in murine models with echocardiographic and histopathological assessment, limits the translation of our findings to whole-organism physiology. On this point, preclinical studies are currently planned in our cardio-oncology group. Third, the downstream molecular effects of PCSK9 silencing beyond the NLRP3 and MYD88 axes remain partially understood; broader omics approaches could better elucidate these mechanisms. These limitations will be systematically addressed in future investigations currently planned by our research group, which aim to validate these findings in preclinical models and extend the analysis to longer treatment durations and combinatorial cardioprotective strategies. In brief, the integration of inclisiran into cardio-oncology practice warrants further investigation, particularly through randomized controlled trials assessing its impact on long-term cardiac outcomes in cancer patients undergoing sequential anthracycline and HER2-targeted therapies [48]. Future studies should also explore its interactions with standard cardioprotective agents, such as beta-blockers, angiotensin-converting enzyme (ACE) inhibitors [49], or sodium-glucose cotransporter-2 inhibitors (SGLT2i) [50], to delineate optimal combination strategies for mitigating chemotherapy-induced cardiac injury. Collectively, these insights highlight the potential of inclisiran as a novel therapeutic approach in cardio-oncology, addressing a critical unmet need in cancer survivorship care.

## 4. Materials and Methods

Doxorubicin and trastuzumab were purchased from Sigma (Milan, Italy). Human induced pluripotent stem cell-derived cardiomyocytes (hiPSC-CMs) were obtained from Ncardia (Leuven, Belgium; Cat. No. CMC-100). Cells were thawed and cultured following the supplier’s recommended protocol. Briefly, cryopreserved hiPSC-CMs were rapidly thawed at 37 °C, centrifuged at 200× *g* for 5 min, and resuspended in pre-warmed cardiomyocyte plating medium (Ncardia Maintenance Medium, Cat. No. CMM-100). Cells were seeded at a density of 50,000 cells/cm^2^ on Matrigel-coated 24-well plates (Corning, Sigma Aldrich, Milan, Italy), and maintained at 37 °C in a humidified incubator with 5% CO_2_. The culture medium was refreshed every 48 h. Cells typically began to exhibit spontaneous contractions within 48 to 72 h post-plating. All experiments were performed between day 7 and day 14 after seeding, a time window during which the cells displayed consistent functional and morphological properties. In brief, in line with the literature [51,52], cardiomyocytes were treated according to this experimental model followed by Pecoraro M. et al. [51]:○Control group (untreated cells): cells were kept in DMEM/F12 10% FBS for 24 h;○Inclisiran group (PCSK9i): cardiomyocytes were treated for 24 h with inclisiran (200 nM);○Doxorubicin + Trastuzumab group (Doxo-Tra): Doxorubicin and trastuzumab sequentially treated cells were treated with doxorubicin (1 µM) for 4 h, and then the medium was replaced with a fresh one, and trastuzumab (200 nM) was added for 20 h.○Doxorubicin + Trastuzumab + Inclisiran: cardiomyocytes were treated as specified in Doxo-Tra group but cells were always co-exposed to inclisiran 200 nM).

### 4.1. Assessment of Cell Survival, Lactate Dehydrogenase, Cytochrome C Release and Caspase-3 Activity During Exposure to Inclisiran Alone or Combined to Anthracyclines and Trastuzumab

Cell viability was assessed using the MTS assay (CellTiter 96^®^ AQueous One Solution Cell Proliferation Assay, Promega, Promega Corporation, Mannheim, Germany, Cat. No. G3582) according to the manufacturer’s protocol. hiPSC-derived cardiomyocytes (Ncardia, Cat. No. CMC-100, 50829 Cologne, Germany) were plated in 96-well plates at a density of 20,000 cells/well on Matrigel-coated surfaces and cultured in Ncardia Maintenance Medium for 7 days prior to treatment. At the end of each treatment described in the previous paragraph, 20 μL of MTS reagent was added directly to each well containing 100 μL of culture medium. Plates were incubated at 37 °C for 2 h, and absorbance was measured at 490 nm using a microplate reader (BioTek Synergy HTX, BioTek Instruments, Inc., Winooski, VT, USA). Cell viability was expressed as a percentage relative to the untreated control group. All conditions were tested in biological triplicates with technical duplicates [53].

Membrane integrity and cytotoxicity were further assessed by quantification of lactate dehydrogenase (LDH) released into the culture supernatant. LDH activity was measured using the Cytotoxicity Detection Kit (LDH) (Roche Applied Science, Indianapolis, IN, USA), following the manufacturer’s instructions. Briefly, cell culture supernatants were collected at the end of each treatment condition and incubated with the LDH reaction mixture. Absorbance was recorded at 490 nm using a multi-well plate spectrophotometer (e.g., BioTek Synergy HTX). LDH release was expressed as a percentage relative to maximal LDH release control wells (positive lysis control), and background values from untreated wells were subtracted accordingly [54]. To assess mitochondrial outer membrane permeabilization (MOMP), cytosolic cytochrome c levels were quantified following subcellular fractionation. After treatment, cells were collected by gentle centrifugation and washed twice with cold phosphate-buffered saline (PBS). Mitochondria-free cytosolic fractions were prepared using the Cell Fractionation Kit (Clontech, Palo Alto, CA, USA), according to the manufacturer’s guidelines. Cells were mechanically lysed by 60 strokes using a pre-chilled Dounce tissue homogenizer (type A pestle) on ice. The homogenate was first centrifuged at 700× *g* for 10 min to remove nuclei and debris. The resulting supernatant was further centrifuged at 10,000× *g* for 25 min at 4 °C to isolate the cytosolic fraction devoid of intact mitochondria. Quantification of cytochrome c in the cytosolic fraction was performed using the Human Cytochrome c ELISA Kit (BioTechne SRL, Milan, Italy). This sandwich ELISA is specifically designed to detect translocated cytochrome c, with a sensitivity of 0.31 ng/mL and a quantification range of 0.6–20 ng/mL. Absorbance was measured at 450 nm using an ELISA plate reader, and concentrations were interpolated from the standard curve generated in parallel [55]. Moreover, for Caspase-3 activity assay, cardiomyocytes were incubated with Caspase-Glo^®^ 3/7 assay reagent (Promega, Madison, WI, USA) for 30 min at 37 °C, in accordance with previously validated protocols assessing apoptosis in models of doxorubicin-induced cardiotoxicity. Luminescence, which is directly proportional to caspase-3/7 activity, was measured using a microplate spectrofluorometer. Results were expressed as pmol/min normalized for mg of protein, in line with the literature [56].

### 4.2. Intracellular Calcium Content and Lipid Peroxidation

Anthracycline-mediated cardiovascular injuries involve high intracellular calcium levels [57]. Intracellular Ca^++^ in human cardiomyocytes was quantified through the fluorescence dye Fluo-3 AM, according to the manufacturer’s protocol. After treatments, cells were loaded with 5 µM Fluo-3 AM at 37 °C for 30 min in the dark, and then washed three times with PBS (pH 7.4) to remove the excess dye. Fluo-3 chelated with Ca^++^ induces fluorescence detected by a spectrofluorometer (excitation/emission wavelengths 488 nm and 525 nm, respectively). Lipid peroxidation in cardiomyocytes was assessed by quantifying two key aldehydic byproducts, malondialdehyde (MDA) and 4-hydroxy-2-nonenal (4-HNE), both indicative of oxidative membrane damage. MDA concentrations were measured using the Thiobarbituric Acid Reactive Substances (TBARS) method, employing the Sigma-Aldrich MDA Assay Kit (MAK085; Sigma-Aldrich, Milan, Italy) according to the manufacturer’s protocol. Briefly, cell lysates were mixed with TBA reagent, incubated at 95 °C for 60 min, and the resultant chromogen was quantified spectrophotometrically at 532 nm. For 4-HNE quantification, an ELISA was performed using the Lipid Peroxidation (4-HNE) Assay Kit (ab238538; Abcam, Milan, Italy) [58]. Samples and standards were incubated in pre-coated 96-well plates, followed by addition of anti-4-HNE antibody and detection reagent, with absorbance measured at 450 nm. All values were normalized to total protein content determined by the BCA assay to account for variations in sample loading.

### 4.3. Mitofusin-2 Intracellular Expression

Expression levels of mitofusin-2 (MFN2), a critical mitochondrial outer-membrane GTPase, were quantified using a sandwich ELISA tailored for human cell lysates [59]. Samples were treated as described in the apoptosis assay; after treatments, cells were treated for intracellular quantification of MFN2 analysis through human MFN2 ELISA Kit (Biomatik, Cat EKU09220, Kitchener, ON, Canada) providing detection between 0.156–10 ng/mL with sensitivity of 0.056 ng/mL. After incubation with biotinylated detection antibody and streptavidin-HRP, TMB substrate was added; the reaction was stopped with sulfuric acid and optical density read at 450 nm. A four-parameter logistic standard curve was generated for concentration interpolation. All assays included blank controls and were performed in biological triplicate. Inter- and intra-assay coefficients of variation were maintained below 12%.

### 4.4. Biomarkers of Cardiotoxicity: Troponin-T, H-FABP and BNP Quantification

Cardiomyocytes were treated as described before. At endpoint, cells were washed with PBS and lysed in RIPA buffer supplemented with protease inhibitors. Lysates were clarified by centrifugation at 10,000× *g* for 10 min, and total protein was quantified by BCA assay. For each biomarker, ELISA kits selected based on assay range and sensitivity were used as follows: Cardiac Troponin-T (cTnT): Quantified using a high-sensitivity TNNT2 sandwich ELISA kit (Assay Genie Human TNNT2 Kit (Dublin, Ireland), sensitivity 9.375 pg/mL; range 15.625–1000 pg/mL). Standards and lysate samples (normalized to 1 mg/mL total protein) were loaded in duplicate to precoated 96-well plates. After 4 h incubation with biotin-conjugated detection antibody and streptavidin-HRP, TMB substrate was added, the reaction halted with acid, and absorbance measured at 450 nm [60]. 

Heart-type Fatty Acid Binding Protein (H-FABP): measured using Abcam’s SimpleStep ELISA Kit (Cambridge, MA, USA; ab243682), sensitivity 2.4 pg/mL and dynamic range 10.9–700 pg/mL. A single incubation with capture and detection antibodies was followed by a single wash, TMB development, and absorbance reading at 450 nm after 90 min.

B-type Natriuretic Peptide (BNP): measured with Abcam’s BNP ELISA Kit (ab193694), with sensitivity of 14 pg/mL and quantification range 14–1000 pg/mL. Samples and standards were incubated in precoated plates, and after sequential application of biotinylated detection antibody, HRP-streptavidin, and TMB, absorbance was read at 450 nm.

In each assay, sample concentrations were interpolated using a four-parameter logistic curve. Controls included blanks and internal positive controls. All assays were performed in biological triplicate; intra- and inter-assay coefficients of variation were maintained below 12%.

### 4.5. F-Actin Expression Assessed Using Confocal Laser Scanning Microscope

Anthracyclines and trastuzumab can reduce F-actin expression in cardiomyocytes [61]. This cytotoxic effect is partly mediated by alterations in membrane lipid composition and associated signaling pathways, which play a critical role in maintaining actin filament organization and cell survival. Lipid metabolism, particularly involving cholesterol and phospholipid homeostasis, has been shown to modulate actin dynamics through interactions with membrane microdomains, small GTPases, and actin-binding proteins [62]. Given the central role of PCSK9 in lipid regulatory pathways and membrane remodeling, we investigated whether inclisiran could preserve F-actin integrity in human iPSC-derived cardiomyocytes exposed to sequential doxorubicin and trastuzumab treatment. Therefore, using a Confocal Laser Scanning Microscope, we investigated the cellular expression of F-actin in cardiomyocytes. HIPSC-CMS cells were cultured as described above. Then, 5  ×  10^3^ cells/well were seeded in a 24-well plate and allowed to grow for 24 h untreated (control) or treated with anthracyclines associated to trastuzumab alone or combined with inclisiran (100 nM) for three days. Then, cells were thoroughly rinsed three times with PBS and fixed with 2.5% glutaraldehyde in PBS for 20 min, as described elsewhere. After washing three times with PBS, cells were permeabilized with 0.1% Triton-X100 in PBS for 10 min and then washed three times with PBS. Subsequently, cells were blocked with 1% BSA in PBS for 20 min. After three washes with PBS, cells were incubated with a Rabbit monoclonal antibody against F-actin (clone ab16502 Abcam) diluted 1:200 in 1% BSA for 1 h. After washing, cells were incubated for 1 h with Goat Anti-Rabbit secondary antibody IgG H&L (FITC) (clone ab6717, AbCam) diluted 1:1000 in 1% BSA. Nuclear staining was obtained through the use of DAPI diluted 1:2000 in PBS for 15–30 min at 37 °C. After washing in PBS, cells were blocked with 1% BSA in PBS for 20 min. A confocal microscope (C1 Nikon) equipped with EZ-C1 software, version 3.90 FreeViewer software. for data acquisition was used (60 × oil immersion objective). Expression of F-actin and the nucleus were imaged using excitation/emission wavelengths of 488/519 nm for F-actin (labeled with Alexa Fluor 488 or FITC) and 358/461 nm for the nucleus (stained with DAPI). F-actin content was quantified using a spectrofluorimetric approach following phalloidin-FITC staining. hiPSC-derived cardiomyocytes were cultured under four experimental conditions: untreated control, PCSK9i alone, DOXO + Anti-HER2, and DOXO + Anti-HER2 + PCSK9i. After treatments, cells were fixed in 4% paraformaldehyde, permeabilized with 0.1% Triton X-100, and stained with phalloidin-FITC (1:1000 dilution) for 30 min at room temperature in the dark. Cells were washed, and fluorescence intensity was measured using a multiwell plate spectrofluorometer (excitation/emission: 495/520 nm). Background signal was subtracted, and readings were normalized to total cell count per well. Results were expressed as mean fluorescence intensity relative to untreated controls.

### 4.6. Quantification of NLRP-3, MyD-88 Cellular Expression

Following experimental treatments, cardiomyocytes were harvested and subjected to lysis in a buffer comprising 50 mM Tris-HCl (pH 7.4), 1 mM EDTA, 100 mM NaCl, 20 mM NaF, 3 mM Na_3_VO_4_, 1 mM PMSF, and a comprehensive protease inhibitor cocktail to preserve protein integrity. Cellular lysates were subsequently centrifuged at 2500 rpm to remove insoluble debris, and the resulting supernatants were collected for downstream analysis [63]. Quantification of intracellular NLRP3 and MyD88 protein levels was performed using enzyme-linked immunosorbent assays (ELISA) specific for human MyD88 (ab171341, Abcam, Milan, Italy) and human NLRP3 (OKEH03368, Aviva Systems Biology, San Diego, CA, USA), in accordance with established protocols [46]. Briefly, 96-well plates pre-coated with monoclonal capture antibodies against either NLRP3 or MyD88 were blocked to prevent nonspecific binding. Subsequently, standards and experimental samples were added and incubated for 1 h to facilitate antigen binding. After rigorous washing, a biotinylated detection antibody specific to the target antigen was introduced, followed by incubation and additional washes to eliminate unbound reagents. An avidin-horseradish peroxidase (HRP) conjugate was then applied, enabling signal amplification. Chromogenic detection was achieved via the addition of TMB substrate, yielding a blue reaction product that was converted to yellow upon the introduction of an acidic stop solution. Optical density was measured at 450 nm, and signal intensity was directly proportional to the concentration of captured analyte. The analytical sensitivity for the MyD88 assay was <10 pg/mL with a dynamic range spanning 156 pg/mL to 10,000 pg/mL, while the NLRP3 assay demonstrated a sensitivity of <0.078 ng/mL and a quantifiable range from 0.156 ng/mL to 10 ng/mL.

### 4.7. Quantitative Profiling of Intracellular Cytokines in Human Cardiomyocytes

Following treatments, human cardiomyocytes were harvested, washed in ice-cold PBS, and lysed in buffer containing 50 mM Tris-HCl (pH 7.4), 150 mM NaCl, 1 mM EDTA, 1 mM PMSF, and a protease inhibitor cocktail. Lysates were clarified by centrifugation at 12,000× *g* for 15 min at 4 °C, and total protein concentration was determined via BCA assay [64]. Cytokine/chemokine levels were quantified using a MILLIPLEX^®^ Human Cytokine/Chemokine/Growth Factor Panel A (48-plex) on a Luminex platform (Bio-Plex), as it includes all analytes of interest: IL-1α, IL-1β, IL-2, IL-4, IL-6, IL-10, IL-12 (p40/p70), IL-17A, IFN-γ, TNF-α, G-CSF, and GM-CSF. In brief, each sample (25 µL containing 200 µg total protein) was incubated with antibody-conjugated magnetic beads for 18 h at 4 °C under gentle agitation. Beads were then washed ×3 with wash buffer and incubated for 1 h at room temperature with the detection antibody cocktail, followed by a 30 min incubation with streptavidin–phycoerythrin. After a final wash, beads were re-suspended in assay buffer and analyzed using a Luminex reader. Analyte concentrations were calculated from a seven-point standard curve using a five-parameter logistic (5-PL) regression model and normalized to total protein, expressed as pg/mg protein. Across samples, intracellular cytokine concentrations ranged from the assay’s lower detection limit (~0.5–10 pg/mL) up to 500 pg/mL, indicating substantial post-treatment expression. Assay sensitivity, dynamic range, and intra-/inter-assay variability were consistent with manufacturer specifications, with CVs generally <10% and quantification limits appropriate for intracellular cytokine detection in cardiomyocytes.

### 4.8. Statistical Analysis

All cell-based assays were performed in triplicate and results are presented as mean  ±  Standard Deviation (SD). Statistical significance was analyzed by Student’s *t* test using Sigmaplot software (Systat Software Inc., version 13.0 (Systat.exe), San Jose, CA, USA). *p*-value  <  0.05 indicates a significant difference between two data values.

## 5. Conclusions

This study provides the first evidence that the PCSK9 inhibitor inclisiran exerts robust anti-inflammatory and cardioprotective effects against cardiotoxicity induced by anthracycline and HER2-targeted therapies. Mechanistically, these effects are mediated through the modulation of the NLRP3 inflammasome and MyD88-dependent signaling pathways. These findings support the potential role of PCSK9 inhibition as a novel therapeutic strategy for the primary prevention of CTRCD, independent of its lipid-lowering properties. The translational relevance of these results underscores the promise of PCSK9i in cardio-oncology, both in preclinical models and potentially in clinical settings for cancer patients at risk of cardiotoxicity.

## Figures and Tables

**Figure 1 ijms-26-06617-f001:**
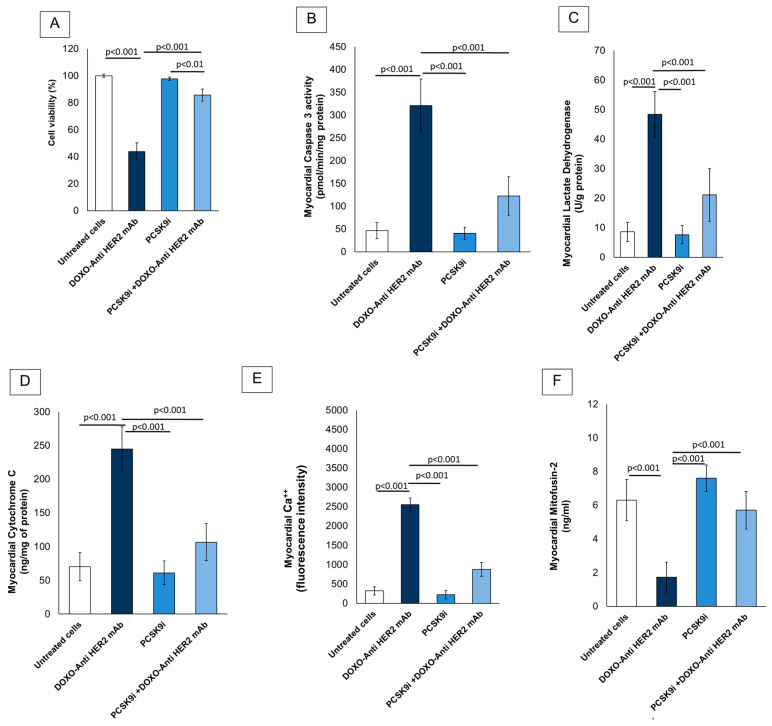
Inclisiran reduces apoptosis and myocardial calcium in cardiomyocytes during anthracycline and trastuzumab therapy in sequential therapy regimen. (**A**) Cell viability (%) of cardiomyocytes determined by MTS assay. (**B**) Myocardial caspase-3 expression levels (pmol/min/mg of protein) in hiPSC-CMS cells as measured using selective ELISA. (**C**) Myocardial LDH levels (U/g of protein) in hiPSC-CMS cells as measured using ELISA method. (**D**) Myocardial cytochrome-c expression in hiPSC-CMS cells (ng/mg of protein). (**E**) Intracellular calcium content in hiPSC-CMS cells, measured by fluorescence intensity (A.U.). (**F**) Myocardial mitofusin-2 expression in cardiomyocytes, expressed in ng/mL. In all experiments, cardiomyocytes were untreated (control) or treated with PCSK9i inclisiran (PSCK9i group) or with anthracyclines and HER-2 mAbs (trastuzumab) in sequential therapy regimen alone (DOXO-Anti HER-2 mAb group) or combined with PCSK9i inclisiran (PCSK9i + DOXO-Anti HER-2 mAb group). (*n* = 3 independent cell culture experiments.) Data are presented as mean ± SD.

**Figure 2 ijms-26-06617-f002:**
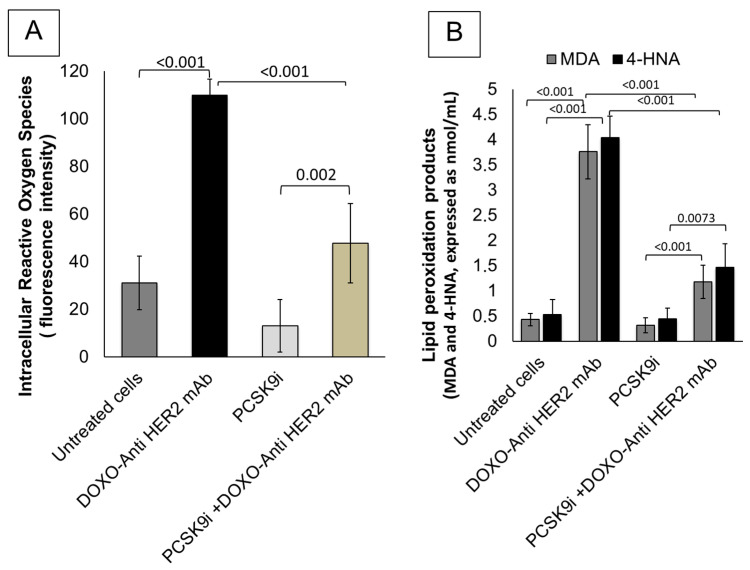
Inclisiran reduces intracellular reactive oxygen species (iROS) and lipid peroxidation products (MDA and 4-HNA) in cardiomyocytes during anthracycline and trastuzumab therapy in sequential therapy regimen. (**A**) Intracellular Reactive Oxygen Species (iROS) in HIPSC-CMS cells as measured using fluorescence intensity. (**B**) Myocardial MDA and 4-HNA levels (nomol/mL) in HIPSC-CMS cells as measured using selective ELISA method. In both experiments, cardiomyocytes were untreated (control) or treated with PCSK9i inclisiran (PSCK9i group) or with anthracyclines and HER-2 mAbs (trastuzumab) in sequential therapy regimen alone (DOXO-Anti HER-2 mAb group) or combined with PCSK9i inclisiran (PCSK9i + DOXO-Anti HER-2 mAb group). (*n* = 3 independent cell culture experiments.) Data are presented as mean ± SD.

**Figure 3 ijms-26-06617-f003:**
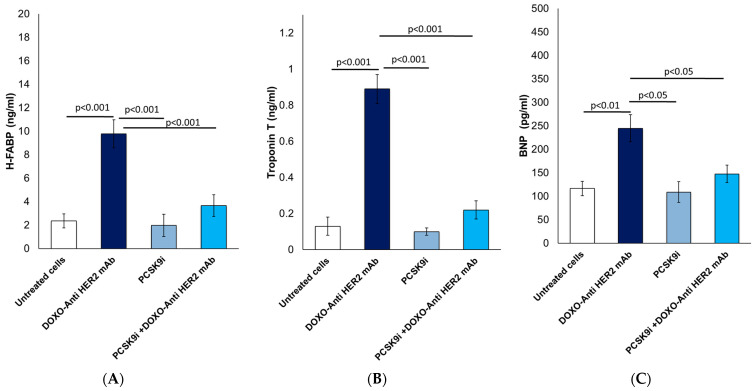
Inclisiran reduces biomarkers of HF, such as H-FABP, troponin-T and BNP in cardiomyocytes during anthracycline and trastuzumab therapy in sequential therapy regimen. (**A**) H-FABP (ng/mL), (**B**) Troponin-T (ng/mL) and (**C**) BNP (pg/mL) were quantified in cytoplasm of human cardiac cells through selective human ELISA methods. In brief, cardiomyocytes were untreated (control) or treated with PCSK9i inclisiran (PSCK9i group) or with anthracyclines and HER-2 mAbs (trastuzumab) in sequential therapy regimen alone (DOXO-Anti HER-2 mAb group) or combined with PCSK9i inclisiran (PCSK9i + DOXO-Anti HER-2 mAb group) (*n* = 3 independent cell culture experiments). Data are presented as mean ± SD.

**Figure 4 ijms-26-06617-f004:**
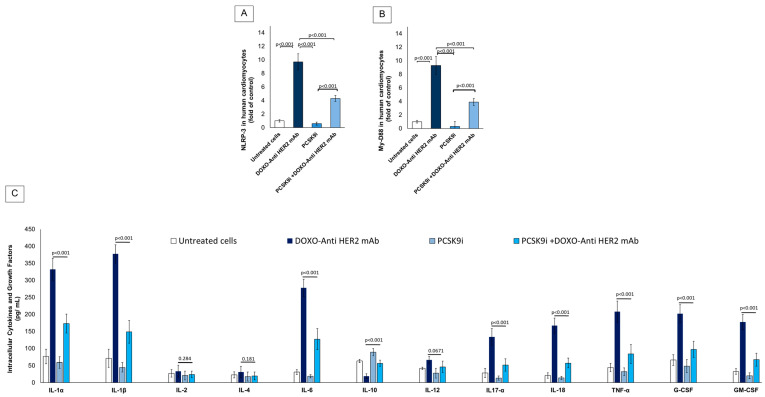
Inclisiran reduces intracellular NLRP-3, MyD88 and pro-inflammatory cytokines in cardiomyocytes during anthracycline and trastuzumab therapy in sequential therapy regimen. (**A**) Intracellular NLRP-3 (fold of control) in hiPSC-CMS cells as measured using selective ELISA method. (**B**) Myocardial MyD-88 levels (fold of control) in HIPSC-CMS cells as measured using selective ELISA method. (**C**) Cytokines and growth factors expression (IL-1α, IL-1β, IL-2, IL-4, IL-6, IL-10, IL-12, IL17-α, IFN-γ, TNF-α, G-CSF, GM-CSF) in pg/mL in HIPSC-CMS cells as measured using selective ELISA method. In all experiments, cardiomyocytes were untreated (control) or treated with PCSK9i inclisiran (PSCK9i group) or with anthracyclines and HER-2 mAbs (trastuzumab) in sequential therapy regimen alone (DOXO-Anti HER-2 mAb group) or combined with PCSK9i inclisiran (PCSK9i + DOXO-Anti HER-2 mAb group). (*n* = 3 independent cell culture experiments.) Data are presented as mean ± SD.

**Figure 5 ijms-26-06617-f005:**
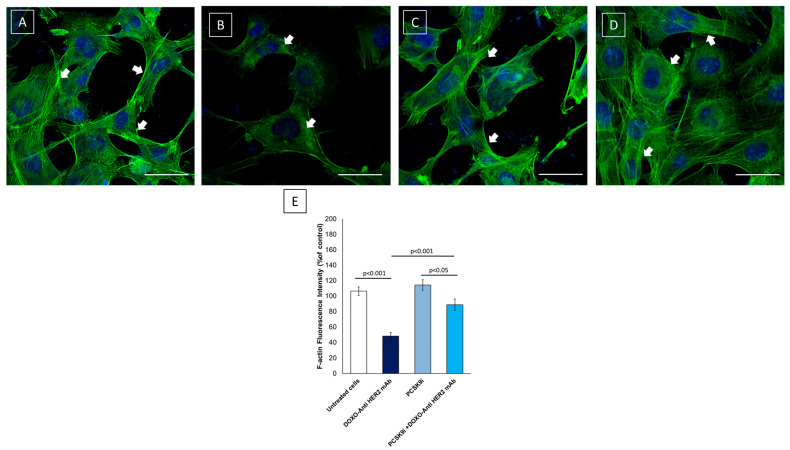
F-actin staining in cardiomyocytes unexposed (**A**) or exposed to anthracyclines (DOXO) and HER2 mAbs (trastuzumab) in sequential therapy alone (**B**) or to inclisiran alone (**C**), or to inclisiran and DOXO-anti- HER2 mAbs in combinatorial regimen (**D**). Green: F-actin staining; blue: nucleus staining through DAPI; scale bar: 50 µm. (**E**) F-actin fluorescence intensity across experimental groups. Data are presented as mean ± SD. White arrows indicates actin filaments in cardiomyocytes.

**Figure 6 ijms-26-06617-f006:**
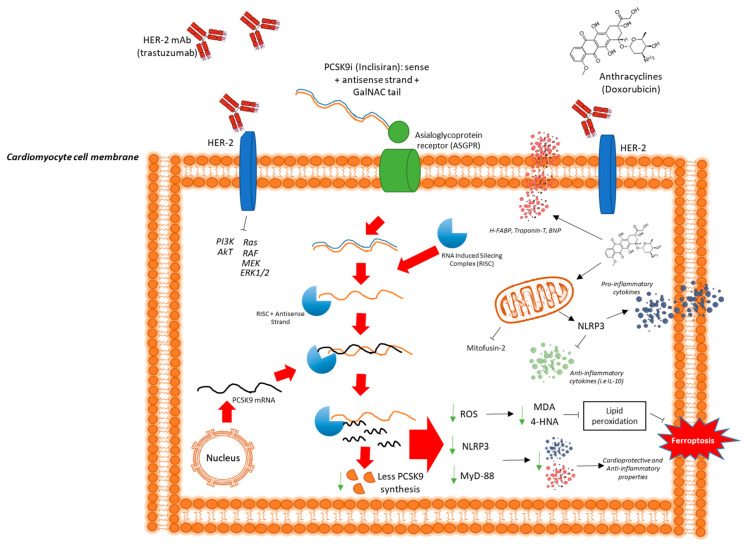
Proposed molecular mechanism by which inclisiran protects cardiomyocytes against doxorubicin and trastuzumab-induced injury. This schematic illustrates the sequential cardiotoxic effects of anthracyclines and trastuzumab and the protective mechanism of inclisiran via RNA interference. At the top left, doxorubicin and trastuzumab bind to their respective targets on the cardiomyocyte membrane. Doxorubicin induces mitochondrial dysfunction and increases reactive oxygen species (ROS) production (red dots), while trastuzumab contributes to oxidative stress and immune-related damage. Inclisiran (green icon) is an siRNA therapeutic targeting PCSK9 mRNA. Upon cellular uptake via receptor-mediated endocytosis, inclisiran enters the cytoplasm and engages the RNA-induced silencing complex (RISC). Within RISC, inclisiran guides the cleavage of PCSK9 mRNA (blue-red duplex), resulting in gene silencing and subsequent reduction of PCSK9 protein levels. This inhibition leads to downstream suppression of pro-inflammatory signaling pathways, including the NLRP3 inflammasome (green specks) and MyD88-dependent cytokine release, reducing mitochondrial damage and oxidative stress. Green arrows indicate decreased expression or activity of oxidative and inflammatory mediators. The reduction in ROS levels and lipid peroxidation (gray clusters) ultimately prevents the initiation of ferroptosis (indicated at bottom right). The diagram also highlights the prevention of cytoskeletal degradation and improved cellular integrity through reduced inflammatory burden. This multifaceted protective mechanism underlines the therapeutic potential of inclisiran in preventing cancer therapy-related cardiac dysfunction (CTRCD).

## Data Availability

The datasets analyzed for this study can be found in the Zenodo repository (https://zenodo.org/records/15638661), accessed on 11 June 2025.

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
