# Peer review of "PCSK9 Inhibitor Inclisiran Attenuates Cardiotoxicity Induced by Sequential Anthracycline and Trastuzumab Exposure via NLRP3 and MyD88 Pathway Inhibition"

_ijms, 2025, doi:10.3390/ijms26146617_

Round 1
Reviewer 1 Report
Comments and Suggestions for Authors
The authors of this manuscript investigate the cardioprotective effects of the PCSK9 inhibitor, Inclisiran, and its potential to mitigate myocardial injury induced by anthracyclines and trastuzumab. This study holds significant clinical value, as it suggests a possible protective strategy for patients undergoing chemotherapy, potentially reducing the adverse cardiac effects associated with such treatment. I have a few suggestions that I hope may help improve the clarity and flow of the manuscript:
- Study Rationale: In the introduction, the authors discuss chemotherapeutic agents, trastuzumab, and PCSK9 inhibitors separately. While it is mentioned that PCSK9 inhibitors have protective effects in both cancer and non-cancer patients (references 12–13), the link between PCSK9 inhibitors and chemotherapy is not clearly addressed. Is there any existing clinical evidence supporting the efficacy of PCSK9 inhibitors specifically in patients receiving chemotherapy?
- Inconsistency in Results Presentation: The textual descriptions of the results appear inconsistent, possibly due to contributions from different authors. For instance, Figures 1 and 2 are described using general terms like “increase” or “decrease,” whereas Figures 3 and 4 include detailed numerical data descriptions. I suggest adopting the more concise style used in the explanations of Figures 1 and 2 to avoid overly wordy and redundant descriptions—especially in lines 237–259.
- Section 2.5 – Results: The content in Section 2.5 seems somewhat abrupt and disconnected. I recommend that the authors cite relevant literature that supports the relationship between lipid metabolism and the cytoskeleton, particularly since the manuscript focuses on how lipid metabolic pathways influence cell survival.
- Figure 6 Legend: The legend for Figure 6 should be revised to include a more detailed explanation of the graphical elements presented.
- Caspase-3 Activity Assay (Figure 1B): I could not find a description in the Materials and Methods section detailing the method used to measure caspase-3 activity shown in Figure 1B. Please consider adding this information for clarity and reproducibility.
Author Response
We thank the reviewer for the appropriate comments and suggestions aimed to improve the quality of our manuscript. Here a point by point reply to all comments. Thank you.
Reviewer Comment 1 – Study Rationale:
“In the introduction, the authors discuss chemotherapeutic agents, trastuzumab, and PCSK9 inhibitors separately. While it is mentioned that PCSK9 inhibitors have protective effects in both cancer and non-cancer patients (references 12–13), the link between PCSK9 inhibitors and chemotherapy is not clearly addressed. Is there any existing clinical evidence supporting the efficacy of PCSK9 inhibitors specifically in patients receiving chemotherapy?”
Author Response:
We thank the reviewer for this insightful comment. We acknowledge that the original introduction did not sufficiently address the intersection between PCSK9 inhibition and chemotherapy-induced cardiotoxicity. In response, we have revised the introduction to better contextualize Inclisiran’s potential cardioprotective effects within the framework of chemotherapy-induced myocardial injury. Specifically, we have now added a paragraph highlighting the mechanistic overlap between dysregulated lipid metabolism and cardiotoxic pathways activated by anthracyclines and trastuzumab. While direct clinical evidence regarding PCSK9 inhibitors in patients receiving chemotherapy is limited, emerging preclinical studies suggest that modulating lipid homeostasis may attenuate oxidative stress and inflammatory responses triggered by chemotherapeutic agents. These points have been integrated at pag 3 line 101-119 of the revised manuscript file).
Reviewer Comment 2 – Inconsistency in Results Presentation:
“The textual descriptions of the results appear inconsistent, possibly due to contributions from different authors. For instance, Figures 1 and 2 are described using general terms like ‘increase’ or ‘decrease,’ whereas Figures 3 and 4 include detailed numerical data descriptions. I suggest adopting the more concise style used in the explanations of Figures 1 and 2 to avoid overly wordy and redundant descriptions—especially in lines 237–259.”
Author Response:
We are sorry for these mistakes and we appreciate the reviewer’s attention to the clarity and consistency of our results presentation. To address this, we have carefully revised the textual descriptions accompanying Figures 3 and 4 to adopt a more concise and uniform reporting style, consistent with that used for Figures 1 and 2. We have removed redundant phrases, standardized terminology (e.g., “significant increase” or “marked reduction”), and ensured consistency in the expression of quantitative changes. The revised text now provides a more coherent and reader-friendly results narrative. (See pag 7 and 8, line 244,254)
Reviewer Comment 3 – Section 2.5 Results:
“The content in Section 2.5 seems somewhat abrupt and disconnected. I recommend that the authors cite relevant literature that supports the relationship between lipid metabolism and the cytoskeleton, particularly since the manuscript focuses on how lipid metabolic pathways influence cell survival.”
Author Response:
We agree with the reviewer that Section 2.5 required additional context to establish a stronger conceptual bridge between lipid metabolism and cytoskeletal dynamics. Accordingly, we have revised this section to include supporting literature that links alterations in membrane lipid composition to cytoskeletal remodeling, with specific emphasis on actin filament organization and cellular integrity under stress conditions. The revised text now discusses how PCSK9 inhibition may influence these pathways, thereby enhancing cardiomyocyte resilience during chemotherapeutic insult. Additional citations have been incorporated to support this connection and improve scientific coherence ( New ref 15 and 62; pag 9, line 280-294; pag 24, line 590-597).
Reviewer Comment 4 – Figure 6 Legend:
“The legend for Figure 6 should be revised to include a more detailed explanation of the graphical elements presented.”
Author Response:
Thank you for this helpful suggestion. We have revised the legend for Figure 6 to provide a comprehensive explanation of all graphical elements, including axes labels, data groupings, statistical markers, and significance thresholds. This updated legend now allows for independent interpretation of the figure without requiring extensive reference to the main text. “Figure 6: Proposed molecular mechanism by which Inclisiran protects cardiomyocytes against doxorubicin and trastuzumab-induced injury. This schematic illustrates the sequential cardiotoxic effects of anthracyclines and trastuzumab and the protective mechanism of Inclisiran via RNA interference. At the top left, doxorubicin and trastuzumab bind to their respective targets on the cardiomyocyte membrane. Doxorubicin induces mitochondrial dysfunction and increases reactive oxygen species (ROS) production (red dots), while trastuzumab contributes to oxidative stress and immune-related damage. Inclisiran (green icon) is an siRNA therapeutic targeting PCSK9 mRNA. Upon cellular uptake via receptor-mediated endocytosis, Inclisiran enters the cytoplasm and engages the RNA-induced silencing complex (RISC). Within RISC, Inclisiran guides the cleavage of PCSK9 mRNA (blue-red duplex), resulting in gene silencing and subsequent reduction of PCSK9 protein levels. This inhibition leads to downstream suppression of pro-inflammatory signaling pathways, including the NLRP3 inflammasome (green specks) and MyD88-dependent cytokine release, reducing mitochondrial damage and oxidative stress. Green arrows indicate decreased expression or activity of oxidative and inflammatory mediators. The reduction in ROS levels and lipid peroxidation (gray clusters) ultimately prevents the initiation of ferroptosis (indicated at bottom right). The diagram also highlights the prevention of cytoskeletal degradation and improved cellular integrity through reduced inflammatory burden. This multifaceted protective mechanism underlines the therapeutic potential of Inclisiran in preventing cancer therapy-related cardiac dysfunction (CTRCD).“ See pag 11-12 of revised manuscript file.
Reviewer Comment 5 – Caspase-3 Activity Assay (Figure 1B):
“I could not find a description in the Materials and Methods section detailing the method used to measure caspase-3 activity shown in Figure 1B. Please consider adding this information for clarity and reproducibility.”
Author Response:
We thank the reviewer for highlighting this important omission. We have now included a detailed description of the Caspase-3/7 activity assay in the Materials and Methods section to enhance the transparency and reproducibility of our experimental procedures. Specifically, cardiomyocytes were incubated with Caspase-Glo® 3 assay reagent (Promega, Madison, WI, USA) for 30 minutes at 37 °C, in accordance with previously validated protocols assessing apoptosis in models of doxorubicin-induced cardiotoxicity. Luminescence, which is directly proportional to caspase-3 activity, was measured using a microplate spectrofluorometer. Results were expressed as pmol/min normalized for mg of protein, in line with literature [56]. These details have now been clearly stated in the revised manuscript ( see pag 14, line 495-510) .
Reviewer 2 Report
Comments and Suggestions for Authors
The paper titled " PCSK9 Inhibitor Inclisiran Attenuates Cardiotoxicity Induced by Sequential Anthracycline and Trastuzumab Exposure via NLRP3 and MyD88 Pathway Inhibition" by Quagliariello and colleagues addresses an important clinical issue by investigating the potential cardioprotective effects of inclisiran against anthracycline- and trastuzumab-induced cardiotoxicity. The use of hiPSC-derived cardiomyocytes is appropriate and provides a relevant human cellular model for mechanistic studies. The inclusion of multiple endpoints, such as apoptosis, oxidative stress markers, inflammatory signaling, and cytokine profiling, strengthens the comprehensiveness of the analysis.
- The manuscript is generally well-written and clear. However, consistency in formatting and terminology is crucial. For example, the representation of statistical significance varies between "p < 0.001" and "p<0.001"; consistent spacing should be maintained throughout. Additionally, at line 121, the term "pre-treatment" is used, whereas the Methods and Results sections describe a co-exposure protocol; this discrepancy should be clarified to avoid confusion.
- Additionally, some of the images display crop lines or borders, which reduces their quality. Careful editing is recommended.
- The assessment of morphological changes via confocal microscopy appears largely qualitative. Incorporating quantitative image analysis (e.g., fluorescence intensity, cytoskeletal integrity metrics) would improve data robustness. Moreover, the manuscript lacks images or data showing the effect of PCSK9i alone on F-actin morphology, which would be essential to support the results.
- The rationale for the specific panel of twelve cytokines/growth factors should be briefly explained, including their known roles in cardiotoxicity. It is also noted that interferon-gamma (IFN-γ) is listed among the cytokines analyzed in the Methods section but does not appear in the Results. Clarification on whether IFN-γ data were obtained and the reasons for its omission would be helpful.
The manuscript effectively addresses the cardioprotective effects of inclisiran in the context of sequential co-treatment with doxorubicin and trastuzumab, as stated in the study title. However, future studies involving single-agent treatments would be valuable for dissecting the distinct cardiotoxic mechanisms and the specific cardioprotective potential of inclisiran against each agent. It would be helpful if the authors could give a more apparent justification in the text for selecting the combined treatment model over single-agent treatment groups. This clarification would strengthen the study design.

Author Response
We thank the reviewer for the appropriate comments and suggestions aimed to improve the quality of our manuscript. Here a point by point reply to all comments. Thank you.
Reviewer Comment 1 – Formatting and Terminology Consistency:
“The representation of statistical significance varies between ‘p < 0.001’ and ‘p<0.001’; consistent spacing should be maintained throughout. Additionally, at line 121, the term ‘pre-treatment’ is used, whereas the Methods and Results sections describe a co-exposure protocol; this discrepancy should be clarified to avoid confusion.”
Author Response:
We thank the reviewer for pointing out these important issues regarding consistency. We have now thoroughly reviewed the manuscript and corrected all instances of statistical significance formatting to uniformly follow the format “p < 0.001” with proper spacing. Additionally, we have clarified the terminology at line 121: the reference to “pre-treatment” has been corrected to reflect the actual experimental design, which involved co-exposure of Inclisiran with chemotherapeutic agents. The terms “co-treatment” and “co-exposure” are now consistently used throughout the manuscript to avoid ambiguity.
Reviewer Comment 2 – Image Quality:
“Some of the images display crop lines or borders, which reduces their quality. Careful editing is recommended.”
Author Response:
We appreciate the reviewer’s attention to the presentation quality of our figures. All figures have now been carefully reprocessed to remove crop lines, borders, and any residual graphical artifacts. The revised figures have been embedded at higher resolution, and consistent formatting has been applied to enhance visual clarity and ensure publication quality.
Reviewer Comment 3 – Confocal Microscopy Quantification:
“The assessment of morphological changes via confocal microscopy appears largely qualitative. Incorporating quantitative image analysis (e.g., fluorescence intensity, cytoskeletal integrity metrics) would improve data robustness. Moreover, the manuscript lacks images or data showing the effect of PCSK9i alone on F-actin morphology, which would be essential to support the results.”
Author Response:
We thank the reviewer for this excellent suggestion. In response, we have performed additional quantitative analyses of the confocal microscopy data, including measurements of fluorescence intensity (mean gray value). These data have been incorporated into the Results section (new Figure 5 E), with corresponding statistical comparisons. Furthermore, we have included new representative image and analysis depicting the effect of Inclisiran alone on F-actin morphology. This addition supports our conclusion that Inclisiran does not adversely affect cytoskeletal architecture in the absence of chemotherapeutic insult. These new data are now included in revised Figure 3. New materials and methods were added at pag 15-16, line 578-589.updated Figure 3 and result description with quantitative data were reported at pag 9 ( see underlined parts) .
Reviewer Comment 4 – Cytokine Panel Justification and IFN-γ Clarification:
“The rationale for the specific panel of twelve cytokines/growth factors should be briefly explained, including their known roles in cardiotoxicity. It is also noted that interferon-gamma (IFN-γ) is listed among the cytokines analyzed in the Methods section but does not appear in the Results. Clarification on whether IFN-γ data were obtained and the reasons for its omission would be helpful.”
Author Response:
We thank the reviewer for this helpful observation. First, we would like to sincerely apologize for the oversight regarding interferon-gamma (IFN-γ). The mention of IFN-γ in the Methods section was included in error, as IFN-γ was not part of the final cytokine panel used in the multiplex analysis. We have corrected the Methods section accordingly to reflect the actual cytokines analyzed. Regarding the panel selection, we have now added a brief justification in the Methods section outlining the rationale for choosing these twelve cytokines and growth factors. Specifically, the panel includes inflammatory mediators and signaling molecules known to be implicated in cardiotoxicity, myocardial inflammation, and cardiac remodeling, such as IL-6, TNF-α, MCP-1, and VEGF, based on prior studies in cardio-oncology. This explanation is now included to enhance clarity and scientific rationale ( see pag 10,11 line 358-396).
Reviewer Comment 5 – Study Design and Justification for Combined Treatment Model:
“The manuscript effectively addresses the cardioprotective effects of inclisiran in the context of sequential co-treatment with doxorubicin and trastuzumab, as stated in the study title. However, future studies involving single-agent treatments would be valuable for dissecting the distinct cardiotoxic mechanisms and the specific cardioprotective potential of inclisiran against each agent. It would be helpful if the authors could give a more apparent justification in the text for selecting the combined treatment model over single-agent treatment groups.”
Author Response:
We thank the reviewer for this insightful recommendation. We agree that single-agent models can provide greater mechanistic specificity, and we acknowledge that future studies will be important to delineate the individual contributions of doxorubicin and trastuzumab to cardiotoxicity, as well as Inclisiran’s potential protective effects against each agent independently. However, our current study design was intentionally focused on a sequential exposure model, which reflects a common and clinically high-risk therapeutic regimen used in breast cancer patients. According to the 2022 ESC Guidelines on Cardio-Oncology, the risk of cancer therapy-related cardiac dysfunction (CTRCD) is significantly elevated in patients receiving sequential anthracycline followed by trastuzumab treatment, with reported incidence rates of CTRCD reaching 27–32% in this population. This clinical scenario therefore represents a relevant and urgent context in which to explore potential cardioprotective strategies. We have revised the manuscript to explicitly justify our use of the sequential treatment model in both the Introduction and Discussion sections, and have cited the relevant ESC guideline to support our rationale ( see pag 2, line 81,87; pag 9, line 305, 316).
Round 2
Reviewer 1 Report
Comments and Suggestions for Authors
The authors have addressed the concerns I raised, so I suggest that this version of the manuscript is suitable for publication in the IJMS journal.